# Cross-sectional online survey of the impact of new tobacco health warnings in Colombia

Sally Adams ![ORCID],[1] Arturo Clavijo ![ORCID],[2] Ricardo Tamayo,[2] Olivia Maynard[3]

¹Department of Psychology, University of Bath, Bath, UK
²Department of Psychology, Universidad Nacional de Colombia, Bogota, Colombia
³School of Experimental Psychology, University of Bristol, Bristol, UK

**Correspondence to**
Dr Sally Adams;
s.adams@bath.ac.uk

## ABSTRACT

**Objectives** This study aimed to explore the impact of a new set of six pictorial warnings introduced in 2018.

**Design and setting** Using a cross-sectional design, we examined awareness of the new warnings among Colombian smokers across two time points of data collection.

**Participants** Adult smokers (≥18 years of age), defined as having smoked at least 100 cigarettes in their lifetime and currently smoking at least one cigarette per week participated at time 1, prior to the introduction of the new health warnings in Colombia in 2018 (n=1985, 72% male), and at time 2, 12 months post introduction (n=1572, 69% male).

**Primary outcomes** At each time, we examined smokers' responses to warnings on packs (negative affect, thinking about warning messages and cognitive elaboration), attitudes toward smoking (perceived likelihood and severity of harm, self-efficacy, response efficacy and quit intentions), knowledge of the health risks of smoking and responses to the new warnings (negative affect, believability, thinking about the harms, reactance and perceived message effectiveness).

**Results** Awareness of the warnings was low, with only 59% of smokers reporting having seen them at time 2. Between times, we observed a reduction in negative affect toward current warnings (p<0.001), reduced thinking about (p<0.001) and cognitive elaboration of the warning message (p<0.001), and an increase in perceived severity of warnings (p<0.001). When asked about the six new health warnings, we found a reduction in negative affect (p<0.07), cognitions related to harm (p<0.01), believability (p<0.03), reactance (p<0.01) and perceived message effectiveness (p<0.02) between times.

**Conclusions** Our data indicate that effectiveness was low prior to the introduction of the new health warnings and at 12 months post introduction. Tobacco control policy should seek to improve exposure to and noticeability of tobacco health warnings in Colombia.

## INTRODUCTION

Every year, over 20 900 Colombian adults die from tobacco-related diseases and despite this, 10% of the population continue to use tobacco each day.[1] Colombia is an emerging market economy, where tobacco use is a growing public health challenge.[2] In response, Colombia has introduced tobacco

## STRENGTHS AND LIMITATIONS OF THIS STUDY

⇒ Our study includes a large sample size, representing smokers from over 100 different areas in Colombia.
⇒ Our work represents some of the first empirical data examining the impact of tobacco control measures such as warnings in Colombia.
⇒ Our study is limited by the use of self-reported data on the perceived effectiveness of health warnings, where it is difficult to determine whether these responses accurately reflect behaviour.
⇒ We were unable to make a direct comparison between participants' responses between study times, as different samples of smokers were recruited. We also observed that many participants failed our attention check, highlighting the limitation of using online self-report data.

control strategies, including restrictions on labelling and packaging of tobacco products, smoke-free places, restrictions to advertising and sales.[3]

Warnings on tobacco products are the most cost-effective tool for educating smokers and non-smokers about the health risks of tobacco use.[4] Pictorial warnings (PWs) are more likely to be noticed than text-only warnings[5–7] and are more effective at educating smokers about the health risk of smoking, increasing cognitions about health risks,[5 8] and at increasing motivation to quit smoking,[5 7 9] while being less vulnerable to the effects of 'wear out'.[8 10] Evidence from low-income and middle-income countries indicates that PWs may be more effective in these countries, being one of the few sources of information about tobacco's risks.[4 11 12] Despite this evidence, there is high variability in the implementation of warnings in low-income and middle-income countries.[13]

PWs were first introduced in Colombia in 2009 and since July 2010 all tobacco products have been required by law to display PWs that cover 30% of the package and are rotated annually. These requirements fall short of the WHO Framework Convention

on Tobacco Control (FCTC) recommendations that warnings cover 50% of the package and that branding is replaced by plain packaging[4]. However, the Colombian Ministry of Health has called for 'local evidence' on the effectiveness of larger warnings and plain packaging, before the implementation of these tobacco control measures. In Colombia, smokers' exposure to warnings may be limited. First, implementation is likely to be slow, particularly in rural areas. Each year when new warnings are introduced, retailers have 12 months to stop selling tobacco products with older warnings (referred to here as the 'phase-in period'). Second, there is a high proportion of Colombian smokers who purchase single sticks, rather than packs, who are, therefore, not exposed to any warnings. Given this uncertainty over the extent to which Colombian smokers are exposed to new warnings, this 1-year cross-sectional study explores smokers' responses to a new set of six PWs covering 30% of the pack, which were introduced in Colombia in 2018.

First, we examined the rate of market saturation and the noticeability of the new warnings among smokers over the implementation period. Given that there is no requirement for retailers to stop selling cigarette packs with the old warnings during the 12-month phase-in period and that many Colombian smokers do not purchase cigarette packs (instead buying single cigarettes), we hypothesised that even a year after implementation, awareness of warnings (a proxy marker of market saturation) will be low (less than 75% of respondents will have seen the warnings). Second, we examined smokers' responses to warnings on packs (negative affect, thinking about warning measures and cognitive elaboration), attitudes toward smoking (perceived likelihood and severity of harm, self-efficacy, response efficacy and quit intentions), knowledge of the health risks of smoking and responses to the new warnings (negative affect, believability, thinking about the harms, reactance and perceived message effectiveness) across the survey times (baseline and 12 months post introduction). Given relatively small changes between the new and old warnings and a low rate of market saturation, we hypothesised that over our assessment period there would be either a small or no increase in negative affect, cognitions related to smoking, knowledge regarding health risks of smoking, perceived severity and likelihood of harm from smoking and quit intentions. Third, we examined smokers' responses to each of the new warnings in terms of negative affect, believability, cognitions related to harm, avoidance, reactance and perceived message effectiveness. Due to wear out (as a result of repeated, although potentially limited exposure) of the warnings, we hypothesised that each of these outcomes would be greatest at baseline as compared with 12 months post exposure. Our hypotheses, methods and analysis plan were pre-registered in our study protocol (osf.io/qbr5v).

## METHODS

### Design
A cross-sectional online survey of Colombian smokers with two times of data collection: one prior to the introduction of the new health warnings and one 12 months post introduction.

### Participants
We recruited adult smokers (≥18 years of age), defined as having smoked at least 100 cigarettes in their lifetime and currently smoking at least one cigarette per week, from the staff and students at the Universidad Nacional de Colombia and from the general public. The research team used existing mailing lists to contact members of the university across eight cities in Colombia and attended student classes to encourage students to complete the surveys. We recruited participants from the general population through social media, word of mouth and posters in public spaces. Due to the exploratory nature of the study, we did not conduct a sample size calculation.

Participants provided their email addresses at baseline. For the second time, participants who completed the baseline time were re-contacted to ask if they would like to participate again. We also sent the survey to individuals who may not have participated in the previous times using the methods described above. At each survey time, participants provided their email address and could choose to enter a prize draw to win one of three iPads.

### Measures and materials
The study team translated all questionnaire measures into Spanish. A professional translator also completed a process of 'back translation'. Following this, we discussed discrepancies and optimal wording and the original Spanish translations were changed. All wording and questions were checked by native Spanish speakers in prestudy piloting.

#### Health warnings
The six warnings introduced on 21 July 2018 depicted six different health-related consequences of tobacco smoking (miscarriage, pancreatic cancer, heart disease, death, anxiety and secondhand smoke) (see online supplemental figure 1).

#### Smoking behaviour
Participants completed smoking behaviour questions to determine eligibility. Demographic measures also included questions on smoker type and smoking heaviness and the first item of the Fagerström Test for Nicotine Dependence[14] (see online supplemental table 1).

#### Awareness of new warnings
Awareness of the new PWs among Colombian smokers was assessed across two times of data collection.

#### Responses to warnings seen on tobacco packs
Responses to warnings seen on packs were assessed using measures of negative affect, thinking about warning message and cognitive elaboration.

### Attitudes to smoking

Attitudes to smoking were assessed using measures of perceived likelihood of harm, perceived severity of harm from smoking, self-efficacy, response efficacy and quit intentions.

### Responses to new warnings

In response to viewing the six new health warnings, participants were asked to complete measures of negative affect, believability, thinking about harms, reactance and perceived message effectiveness.

### Knowledge of health risks

We assessed knowledge of the six specific health risks presented on the new health warnings.

### Procedure

The first time of data collection occurred between 21 May 2018 and 12 July 2018, prior to 21 July 2018, the date at which new warnings could start appearing on the market. The second time of data collection took place between 5 June 2019 and 1 August 2019. All procedures were identical at the two times.

Participants accessed the study via Qualtrics and were first shown an information statement before completing a tick-box consent page. All text and questions were presented in Spanish. Participants completed screening questions and those who were ineligible were taken to the end of the study. Eligible participants completed demographic questions, responses to warnings on packs, knowledge of health risks and attitudes toward smoking. An attention check item was included in this section (please select the strongly agree option to let us know that you read all of the survey instructions). Each of the six new warnings were presented individually on screen in a randomised order, and participants completed questions about each warning. After reading a debriefing page, participants who wished to be entered into the prize draw were directed to a second survey. Participants who participated in the first time were re-contacted at the second time to ask if they would like to participate again.

### Data analysis

To examine market spread of new warnings over the implementation period, we calculated percentages of participants in the two times who reported being aware of seeing each of the six new warnings. In exploratory analyses, we examined how warning awareness differs between smoking status (daily and weekly smokers) and smoker type (buys cigarette packs, buys singles, buys both).

To examine the impact of the new warnings, we compared responses to warnings on packs, knowledge of health risks and attitudes to smoking between time 1 and time 2 using independent samples' t-tests. We ran these analyses both including and excluding participants who were defined as unexposed to the new warnings at time 2. We defined participants as 'unexposed' if they reported not seeing (or being unsure whether they had seen) 3 or more of the 6 new warnings at time 2.

To examine smokers' responses to each of the new warnings, we compared negative affect, believability, thinking about the harms, reactance and perceived message effectiveness for each of the six warnings at time 1 and time 2 using independent samples' t-tests.

The internal consistency of the latent constructs (eg, negative affect and cognitive elaboration) were assessed using Cronbach's alpha. All latent constructs achieved acceptable to good internal consistency and latent variables were calculated by taking a mean of the individual items (see online supplemental table 2).

We conducted analyses using IBM SPSS Statistics V.22. The data that form the basis of the results were generated by the authors of this manuscript (dataset).[15]

### Patient and public involvement

It was not appropriate or possible to involve patients or the public in the design, or conduct, or reporting, or dissemination plans of our research.

## RESULTS

### Data screening and participant characteristics

Data were examined prior to analysis and participants who completed the survey after the closing date (n=6 in time 1) or failed the attention check (n=1866 in time 1 and n=988 in time 2) were removed from further analyses. Participant characteristics are presented by data collection time in table 1.

### Awareness of new warnings

As expected, participants' awareness of new tobacco health warnings (ie, participants reporting that they had seen the warning) increased from time 1 (33%) to time 2 (59%; see figure 1). Note that the new warnings had not been introduced on packs by time 1, so the 33% who reported seeing them were mistaken. Participants mistakenly reporting that they had seen the new warnings at time 1 was highest for the secondhand smoke (47%), miscarriage (45%) and heart disease (46%) warnings and lowest for the death (25%), pancreatic cancer (21%) and anxiety (18%) warnings.

Awareness of the new health warnings at time 2 was higher for smokers who reported buying only cigarette packs or both pack and single cigarettes, compared with those who bought only single cigarettes. Awareness of the new health warnings at time 2 was also higher for daily smokers compared with weekly smokers (see online supplemental figures 2 and 3).

### Responses to warnings seen on tobacco packs

As listed in table 2, there was evidence of a decrease in negative affect, thinking about warning messages and cognitive elaboration between time 1 and time 2. These findings did not differ when including only participants defined as 'exposed' at time 2.

### Attitudes to smoking

There was strong evidence of an increase in perceived severity between time 1 and time 2 (table 2), although

**Table 1** Participant characteristics by data collection time

| | n=1985 | n=1572 | P value (CIs) |
|---|---|---|---|
| Sex (% male) | 72 | 69 | N/A |
| Age (years) | 26 (9) | 26 (8) | 0.03 (−1.098 to 0.049) |
| Location (% Bogota) | 49 | 67 | <0.001 |
| Smoking frequency | | | |
| Daily smokers (%) | 65 | 62 | 0.06 |
| Cigarettes per day (daily smokers) | 7 (6) | 7 (5) | N/A |
| Weekly smokers (%) | 35 | 38 | 0.06 |
| Cigarettes per week (weekly smokers) | 10 (10) | 9 (9) | 0.09 |
| How many days a week do you smoke (weekly smokers)? | 3 (1) | 3 (1) | 0.42 |
| Cigarettes per day (weekly smokers) | 3 (3) | 3 (3) | 0.38 |
| Type of packs bought | | | |
| Buys cigarette packs (%) | 38 | 31 | <0.001 |
| Buys single cigarettes (%) | 22 | 26 | N/A |
| Buys both packs and single cigarettes (%) | 39 | 41 | 0.21 |
| Dependence | | | |
| Smokes within first 60 min of waking (%) | 27 | 24 | 0.02 |

Values are %, except for age, cigarettes per day, cigarettes per week and smoking days per week—mean (SD). P values represent differences in variables between times (t-test/$\chi^2$ test).

this difference was not evident after including only participants defined as exposed to the warnings at time 2. The analysis including only participants exposed at time 2 also indicated strong evidence of a decrease in self-efficacy and quit intention between times. There was no evidence for any other differences.

### Knowledge of health risks

There was strong evidence of an increase in mean knowledge of health risks between times across all warnings, when including only participants defined as exposed at time 2 (see table 2). For individual warnings, there was evidence of a small but statistically meaningful increase in knowledge for the health risks: 'smoking causes

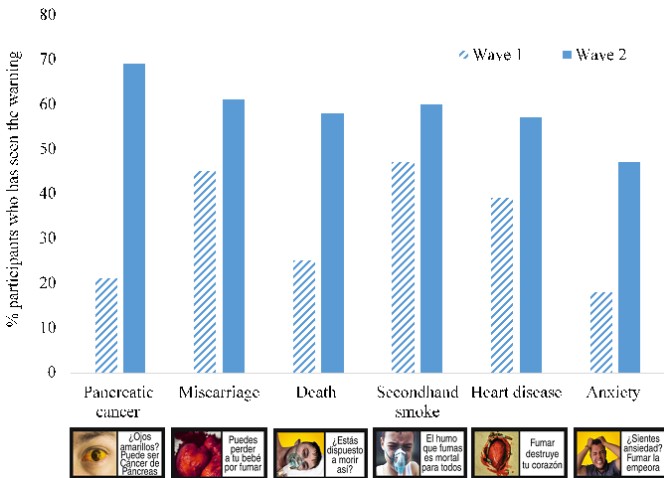

**Figure 1** Percentage of participants who report being aware of having seen the new warnings at waves 1 and 2.

pancreatic cancer' and 'smoking causes anxiety' between times. The analysis including only participants exposed to the warnings at time 2 indicated that there was also evidence of an increase between times in knowledge for the health risk: 'smoking while pregnant could cause you to lose your baby'. These data were highly skewed, however, with high rates of participants giving scores of either 0 (strongly disagree) or 100 (strongly agree) for each knowledge question (see table 3).

### Responses to new warnings

Differences in responses to the six new health warnings are listed in table 4.

For the health warning 'miscarriage', there was strong evidence of a decrease in negative affect, cognitions related to harm, believability and perceived message effectiveness between times. For the health warning 'pancreatic cancer', there was strong evidence of a decrease in cognitions related to harm and reactance and an increase in believability between times. For the health warning 'anxiety', there was no evidence of any differences in response to the warning. For the health warning 'heart disease', there was strong evidence of a decrease in negative affect, cognitions related to harm and perceived message effectiveness between waves. For the health warning 'secondhand smoke', there was strong evidence of a decrease in negative affect, cognitions related to harm, believability and perceived message effectiveness between times. Finally, for the health warning 'death', there was strong evidence of a decrease in negative affect, cognitions related to harm and perceived message

**Table 2** Responses to tobacco health warnings and attitudes to smoking by data collection time (for all participants and excluding participants unexposed to new warnings at time 2)

| | Time 1 (n=1985) | Time 1 vs time 2—including all participants at Time 2 | | | | Time 1 vs time 2—only including participants exposed to new warnings at time 2 | | | |
| --- | --- | --- | --- | --- | --- | --- | --- | --- | --- |
| | | Time 2 (n=1572) | T value | Df* | P value | Time 2 (n=910) | T | Df* | P value |
| **Responses to warnings on packs** | | | | | | | | | |
| Negative affect | 41 (27) | 38 (26) | 3.03 | 3411 | <0.001 | 38 (26) | 3.35 | 2893 | <0.001 |
| Thinking about warning messages | 31 (26) | 28 (25) | 3.28 | 3454 | <0.001 | 27 (25) | 3.28 | 1880 | <0.001 |
| Cognitive elaboration | 46 (31) | 42 (29) | 3.54 | 3432 | <0.001 | 42 (29) | 3.59 | 1867 | <0.001 |
| Knowledge of health risks (overall) | 73 (17) | 74 (16) | −1.78 | 3443 | 0.08 | 76 (17) | −3.91 | 2893 | <0.001 |
| Lose your baby† | 85 (22) | 85 (21) | −0.59 | 3470 | 0.56 | 87 (20) | −2.24 | 1979 | 0.03 |
| Smoking causes pancreatic cancer | 68 (29) | 70 (27) | −2.2 | 3555 | 0.03 | 72 (28) | −3.53 | 2893 | <0.001 |
| Secondhand smoke is deadly | 74 (27) | 73 (26) | 1.2 | 3555 | 0.23 | 74 (26) | 0.41 | 2893 | 0.68 |
| Smoking damages your heart | 83 (21) | 83 (20) | −0.03 | 3555 | 0.98 | 84 (20) | −0.9 | 2893 | 0.37 |
| Smoking causes anxiety | 60 (36) | 64 (34) | −3.41 | 3471 | <0.001 | 67 (33) | −5.67 | 1948 | <0.001 |
| Smoking causes a slow and painful death | 68 (30) | 68 (28) | −0.59 | 3460 | 0.56 | 70 (28) | −1.93 | 1873 | 0.05 |
| **Attitudes to smoking** | | | | | | | | | |
| Perceived likelihood of harm | 78 (19) | 77 (17) | 0.61 | 3467 | 0.54 | 78 (17) | −0.3 | 1887 | 0.76 |
| Perceived severity of smoking harms | 87 (17) | 89 (15) | −3.97 | 3501 | <0.001 | 88 (16) | −1.73 | 1815 | 0.08 |
| Self-efficacy | 69 (32) | 67 (32) | 1.65 | 3555 | 0.1 | 64 (33) | 3.73 | 1708 | <0.001 |
| Response efficacy | 83 (24) | 84 (22) | −0.54 | 3487 | 0.59 | 84 (22) | −0.32 | 1899 | 0.75 |
| Quit intention | 60 (30) | 59 (30) | 1.61 | 3554 | 0.11 | 56 (31) | 3.9 | 2892 | <0.001 |

Vales are mean (SD), *t*-score, df and *p*-values.
*Df differ according to equality of variances of variables.
†Full wording is 'Smoking while pregnant could cause you to lose your baby'.

**Table 3** Knowledge of health risks (percentage of participants giving a score of 0 or 100) by data collection time (for all participants and excluding participants unexposed to new warnings at time 2)

| | Time 1 (n=1985) | | Time 1 vs time 2—including all participants at time 2 | | Time 1 vs time 2—only including participants exposed to new warnings at time 2 | |
| --- | --- | --- | --- | --- | --- | --- |
| | | | Time 2 (n=1572) | | Time 2 | |
| | % giving a score of 0 | % giving a score of 100 | % giving a score of 0 | % giving a score of 100 | % giving a score of 0 | % giving a score of 100 |
| Warning 1: 'pancreatic cancer' | 4 | 24 | 3 | 25 | 3 | 28 |
| Warning 2: 'miscarriage' | 1 | 50 | 1 | 48 | 1 | 51 |
| Warning 3: 'death' | 5 | 25 | 3 | 23 | 3 | 25 |
| Warning 4: 'secondhand smoke' | 2 | 31 | 1 | 27 | 1 | 30 |
| Warning 5: 'heart disease' | 1 | 41 | 1 | 39 | 1 | 42 |
| Warning 6: 'anxiety' | 11 | 25 | 7 | 25 | 6 | 28 |

Values are % of participants giving a score of 0 ('strongly disagree') or 100 ('strongly agree') for the 6 new warnings.

**Table 4** Responses to the new tobacco health warnings by data collection time

|  | Time 1 (n=1985) | Time 2 (n=1572) | T value | Df* | P value |
|---|---|---|---|---|---|
| Warning 1: 'pancreatic cancer' |  |  |  |  |  |
| Negative affect | 47 (30) | 45 (28) | 1.79 | 3093 | 0.07 |
| Cognitions related to harm | 59 (32) | 57 (32) | 2.55 | 3134 | 0.01 |
| Believability | 57 (32) | 60 (31) | −2.97 | 3076 | <0.001 |
| Reactance | 41 (25) | 39 (25) | 2.52 | 3134 | 0.01 |
| Perceived message effectiveness | 48 (27) | 47 (26) | 1.52 | 3134 | 0.13 |
| Warning 2: 'miscarriage' |  |  |  |  |  |
| Negative affect | 50 (30) | 44 (29) | 5.57 | 3128 | <0.001 |
| Cognitions related to harm | 59 (34) | 51 (35) | 6.16 | 2984 | <0.001 |
| Believability | 67 (31) | 64 (31) | 2.92 | 3128 | <0.001 |
| Reactance | 43 (27) | 42 (27) | 0.84 | 3128 | 0.4 |
| Perceived message effectiveness | 51 (27) | 47 (27) | 4.1 | 3128 | <0.001 |
| Warning 3: 'death' |  |  |  |  |  |
| Negative affect | 55 (30) | 51 (29) | 3.52 | 3136 | <0.001 |
| Cognitions related to harm | 67 (31) | 64 (31) | 2.75 | 3136 | 0.01 |
| Believability | 68 (30) | 66 (30) | 1.5 | 3136 | 0.13 |
| Reactance | 42 (27) | 41 (25) | 0.4 | 3096 | 0.69 |
| Perceived message effectiveness | 54 (26) | 52 (27) | 2.31 | 3136 | 0.02 |
| Warning 4: 'secondhand smoke' |  |  |  |  |  |
| Negative affect | 46 (29) | 42 (28) | 3.52 | 3123 | <0.001 |
| Cognitions related to harm | 61 (33) | 55 (33) | 4.95 | 3123 | <0.001 |
| Believability | 69 (30) | 66 (30) | 2.19 | 3123 | 0.03 |
| Reactance | 40 (25) | 40 (25) | −0.19 | 3123 | 0.85 |
| Perceived message effectiveness | 48 (26) | 45 (26) | 3.39 | 3123 | <0.001 |
| Warning 5: 'heart disease' |  |  |  |  |  |
| Negative affect | 49 (30) | 45 (28) | 3.33 | 3068 | <0.001 |
| Cognitions related to harm | 64 (31) | 60 (31) | 3.41 | 3112 | <0.001 |
| Believability | 66 (30) | 65 (30) | 1.6 | 3112 | 0.11 |
| Reactance | 38 (25) | 37 (24) | 1.51 | 3058 | 0.13 |
| Perceived message effectiveness | 49 (26) | 46 (25) | 3.47 | 3112 | <0.001 |
| Warning 6: 'anxiety' |  |  |  |  |  |
| Negative affect | 30 (28) | 29 (26) | 1.2 | 3089 | 0.23 |
| Cognitions related to harm | 42 (35) | 41 (33) | 1.04 | 3084 | 0.3 |
| Believability | 51 (34) | 53 (34) | −1.4 | 3131 | 0.16 |
| Reactance | 41 (25) | 40 (25) | 1.14 | 3131 | 0.26 |
| Perceived message effectiveness | 36 (25) | 35 (25) | 0.93 | 3131 | 0.35 |

Values are mean (SD), t score, df and p values.
*Df differ according to equality of variances of variables.

effectiveness between times. There was no evidence for any other differences.

## DISCUSSION

Every year since 2010, Colombian smokers have been faced with new health warnings covering 30% of their cigarette packs. Using a cross-sectional design, we examined how smokers responded to the new set of warnings introduced in 2018 and find that they had relatively limited impact on knowledge of risk and attitudes toward smoking. We do, however, find some interesting differences between the warnings themselves, and together these findings have important implications for health warning design and implementation in Colombia and worldwide.

As hypothesised, we found that awareness of the warnings was low. Across the six warnings, the average percentage of participants reporting having seen them at time 2 was 59%. This is particularly low, given that even at time 1, 33% of participants incorrectly reported having seen the warnings. Given this low level of awareness, it is unsurprising that we see either no changes or relatively small changes in our measures of responses to warnings on packs, overall knowledge of health risks and attitudes to smoking between time 1 and time 2—these are outcomes we would expect to improve, had the new warnings made an impact on smokers. Although the differences were small, contrary to our hypothesis, participants reported lower negative affect, thinking about the warnings and cognitive elaboration of the warnings on their cigarette packs at time 2. Even when we only include those participants we defined as 'exposed' to the new warnings at time 2, these unexpected effects persisted. We saw little to no change in attitudes to smoking between times, although there was some evidence that participants at time 2 thought that the severity of smoking harms was greater, and they had greater knowledge of the health risks of smoking. However, again, contrary to expectations, when we only examined these effects among participants who we defined as exposed to the new warnings at time 2, self-efficacy and quit intentions were lower at time 2.

Together, these findings suggest that the new warnings had relatively little impact on smokers. We consider there to be a number of complementary explanations for this. First, although time 2 was conducted 12 months after the new warnings were permitted for sale, it is likely that they did not become commonplace for some time after that, as there is no legal requirement for retailers to stop selling packs with older tobacco warnings until 12 months after the implementation period. It is possible that this slow phase-in diminished warnings' impact and meant that even a year after they (in theory) started being available on packs for sale, few participants recalled having seen them. Second, low awareness of warnings is also likely related to the extent to which smokers engage with cigarette packs. Despite being prohibited by law, many Colombian smokers purchase single cigarettes[16] and, therefore, may not be exposed to health warnings at all. Indeed, we find that awareness of the new warnings is greatest for those who smoke daily and those who buy packs.

We do, however, find some interesting differences between the warnings themselves. As might be expected, we see the highest levels of participants incorrectly reporting at time 1 that they had seen the new warnings for those with themes used in previous years ('miscarriage', 'heart disease' and 'secondhand smoke' were used most recently in 2017 and 'death' was used in 2015; see online supplemental table 3). For these four 'familiar' warnings (similar theme, but different wording and pictorial), we see high levels of believability and knowledge of these health risks at both times and only a small proportion of participants reporting that they 'strongly disagreed' that smoking caused these health risks.

However, for all four of these familiar warnings, we observe reductions between times in almost all specific responses to them (eg, perceived message effectiveness and negative affect). These data may provide support for previous work, which has observed a declining impact of warnings (ie, 'wear out') over time.[7 17–20] However, wear out assumes that the warnings were initially effective but become less effective with time. The six warnings tested here received relatively low scores on all measures at time 1 and, although statistically significant, the observed declining impact of warnings at time 2 was small and may not reflect a meaningful change. These low scores for the warnings may suggest that they are not effective at eliciting responses such as negative affect, cognitions related to harm, believability and perceived message effectiveness, and it is possible that their effectiveness may be limited by the theme or disease depicted in the warning.

There are two warning themes which were new on Colombian cigarette packs in 2018, 'pancreatic cancer' and 'anxiety'. As expected, for these novel warnings, we saw a much smaller percentage of participants incorrectly reporting that they had seen them at time 1 than the 4 familiar warnings. However, interestingly, at time 2, we saw divergent levels of awareness for these 2 warnings. The largest percentage of participants reported having seen the pancreatic cancer warning of all warnings at time 2, but the lowest percentage of all participants reported having seen the anxiety warning. We suggest this may be due to the pictorial used, with the pancreatic cancer warning displaying a close-up of a yellow eye, which is arguably novel and memorable, while the anxiety warning depicted a man pulling at his hair. Previous research has suggested that more 'severe' or grotesque warnings increase smokers' intentions to quit and are rated as more believable and effective than less severe warnings.[21 22]

Knowledge of the risk of smoking for both pancreatic cancer and anxiety was low at time 1 and at this time-point, we saw the highest percentages of participants reporting that they 'strongly disagreed' that smoking caused these conditions. The inclusion of these new warning messages reflects the priority for the Colombian Ministry of Health to inform the public about 'lesser known' smoking health harms. We saw increased knowledge of these two health risks at time 2, and fewer participants selecting 'strongly disagree' particularly for anxiety, although knowledge was still lower than for most other health risks. Despite this, these increases suggest that novel warnings can impact smokers' knowledge of smoking risks. We also saw that the anxiety warning had the lowest response scores at both times compared with all other warnings (particularly negative affect, believability, perceived message effectiveness and cognitions related to harm). We suggest that these low scores, combined with low 'knowledge' scores, are in line with the inconsistent cross-sectional evidence regarding the relationship between smoking and anxiety,[23] and evidence from Mendelian randomisation studies which find no causal relationship between

smoking and the onset of anxiety.[24] We suggest that it is important to consider the veracity of the health risk claims depicted in health warnings and we raise the important question of whether including health risks which are not supported by the evidence might reduce the overall impact of warnings.

Warnings have repeatedly been shown to be a cost-effective tool for communicating the health risks of smoking to smokers' and non-smokers.[4] For low-income and middle-income countries, warnings are proposed to be even more effective.[4] However, for this to occur, smokers must be exposed to warnings, where increased noticeability may improve warning effectiveness. Our data suggest that strategies for increasing exposure and noticeability of warnings are required in Colombia. Currently, warnings in Colombia cover 30% of the pack, with the FCTC recommending that warnings should cover at least 50%. Noticeability and subsequent effectiveness of warnings may be improved by the introduction of larger pictorial health warnings.[25 26] Additionally, as a large proportion of smokers purchase single cigarettes rather than packs, including health warning messages on individual cigarette sticks represents a novel and effective method for reducing tobacco use.[27–29]

Our study has numerous strengths, including a large sample size, representing smokers from over 100 different areas in Colombia. Additionally, our work represents some of the first empirical data examining the impact of tobacco control measures such as warnings in Colombia. However, our findings should also be considered in light of the following limitations. Our study is limited by the use of self-reported data on the perceived effectiveness of health warnings, where it is difficult to determine whether these responses accurately reflect behaviour. We were unable to make a direct comparison between participants' responses between study times, as different samples of smokers were recruited. We also observed that many participants failed our attention check, highlighting the limitation of using online self-report data. Our participants may not be representative of the general population in Colombia and in particular smokers. Internet penetration is estimated at 65% in Colombia,[30] where individuals without internet access would not have been able to participate in our survey. Furthermore, our sample of smokers were on average younger, more educated and more likely to be male compared with Colombian smokers reported in previous research.[31] Therefore, our sample may limit the generalisability of our findings to all smokers in Colombia, where further research on the effectiveness of tobacco health warnings for all Colombian smokers is warranted.

## CONCLUSIONS

Health warnings represent a cost-effective tool for communicating the health risks of smoking in low-income and middle-income countries. However, our data suggest that current warnings have a limited impact on knowledge of risk and attitudes toward smoking. Future tobacco control policy should seek to improve the noticeability of and exposure to tobacco health warnings in Colombia. Strategies may include consideration of veracity of the health risk claims depicted in health warnings, the inclusion of health warnings on single cigarette sticks and increased warning size on tobacco products.

**Contributors** Conception and design of the work: SA, RT, AC and OM. Analysis: SA and OM. Interpretation of data for the work: all authors. Drafting the work: SA and OM. Revising it critically and final approval of the version to be published: all authors. SA is acting as guarantor.

**Funding** This research was funded by an ESRC New Investigator's Award, awarded to OM (ES/R003424/1), and a University of Bristol Global Challenges Research Ad Hoc Fund.

**Competing interests** RT and AC were previously hired by the Colombian Ministry of Health to design the six warnings used here. The Ministry of Health had no role in the design of the study.

**Patient and public involvement** Patients and/or the public were not involved in the design, or conduct, or reporting, or dissemination plans of this research.

**Patient consent for publication** Not applicable.

**Ethics approval** This study involves human participants and was approved by the ethics committee of Faculty of Human Sciences, Universidad Nacional de Colombia (ID: B.VDIECH-024-18). Participants gave informed consent to participate in the study before taking part.

**Provenance and peer review** Not commissioned; externally peer reviewed.

**Data availability statement** Data are available in a public, open access repository. The data that form the basis of the results presented here are available from the University of Bristol's data repository, data.bris, at https://doi.org/10.5523/bris. 3fscvnkvn2s2e2ev9vint0lbj3. These data were generated by the authors of this manuscript [dataset].

**ORCID iDs**
Sally Adams http://orcid.org/0000-0001-8398-4460
Arturo Clavijo http://orcid.org/0000-0002-8283-9517

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
