## [Reviewer comments · BMJ Open]

ARTICLE DETAILS

TITLE (PROVISIONAL)	A cross-sectional online survey of the impact of new tobacco health warnings in Colombia.
AUTHORS	Adams, Sally; Clavijo, Arturo; Tamayo, Ricardo; Maynard, Olivia

VERSION 1 – REVIEW

REVIEWER	Gendall, Philip University of Otago, Public Health
REVIEW RETURNED	11-Oct-2021

GENERAL COMMENTS	Your paper has a rather long list of potential limitations that inevitably reduce the robustness of your findings and conclusions: One third of the Wave 1 participants reported seeing the new pictorial warnings before they had been introduced, the study relies on self-reports from a survey in which a large number of respondents failed an attention check (though these respondents were removed from the analyses), the two samples contained different respondents and were convenience samples, and though the characteristics of both samples were similar, it is not clear if the achieved samples were representative of Colombian smokers. Furthermore, I believe your conclusion that ‘wear-out’ is a major problem is not necessarily suggested by your results. Despite these limitations, the research itself is well designed, the data appropriately analysed, the paper is generally well written and you acknowledge your study’s limitations. The paper would be of interest to those involved in tobacco control, particularly in low-to-middle income countries where less research of this type has been conducted. With some minor revisions I believe the paper would be suitable for publication. Your surveys were conducted on line. I believe the level of internet penetration in Colombia is only 65%, so there is potential coverage error in your samples. Also, your convenience sampling methods are likely to have produced non-representative samples of those with access to the internet. Consequently, it is not clear how representative your samples were of Colombian smokers in general (for example, I suspect your respondents would be better educated than the general population). Ideally, you would include in Table 1 the characteristics of Colombian smokers in general and comment on any significant differences between your samples and the smoker population. I understand the logic of your pre-post design and appreciate that you have separate analyses for all of your wave 2 respondents and only for those who were not ‘exposed’ in wave 1. However, I wonder how much value there actually is in comparing your
---

respondents' evaluations a year apart. The significant differences in mean scores in Tables 2 and 4 are almost all less than five percentage points and they are highly significant simply because of the very large sample sizes concerned. Arguably, many of these differences are not practically important, and your between-subjects design means it is difficult to know if these differences are 'real' or simply a reflection of differences in your samples.

These between-sample comparisons are necessary to test your 'wear-out' hypothesis, but as I explain below, I believe wear-out is only one plausible explanation for your findings and not one that can be proven by these findings. The deficiencies in the Colombian warnings are well illustrated by the results of your second survey and support your conclusions that something needs to be done to improve the notability, exposure and effectiveness of these warnings. However, if you want to continue to argue that your study is a test of 'wear out', I think you need to clearly explain why 'wear-out' is the inevitable cause of the reduction in negative effect, etc. in wave 2 and acknowledge that statistically significant differences may not be practically important. In other words, I think you should be a little more circumspect in the interpretation of your results.

Regardless of this, I am not sure your results support your conclusion that wear-out is a major cause of the reduced effectiveness of the new warnings you observed, or that larger warnings would reduce this wear-out effect. Colombian smokers may well have become desensitised to warnings about miscarriage, second hand smoke and heart attack. However, as I have already mentioned, your between-subjects design makes it difficult to be certain about the apparent decline in the effectiveness of these warnings or its causes and you do not explain how larger warnings would reduce wear-out (if this is what is occurring). Furthermore, I believe there are other equally plausible conclusions that could be drawn from your results.

As you point out, the Colombian warnings cover only 30% of the pack, not the 50% recommended by the WHO, so the warnings are relatively small and arguably easy to ignore or overlook. The images themselves seem rather benign to me, compared to some of the gruesome health images used in other countries. Also, four of the six images are unlikely to be very convincing in my opinion: the anxiety and pancreatic cancer images are not obviously associated with the consequences of smoking, and the death and second hand smoke images are very similar and not particularly compelling. I appreciate that the design and choice of warning images is not an easy task, but I think emphasising wear-out ignores the real problem with the Colombian warnings; namely, that they are too small and not well executed.

A few minor points:

A brief explanation of your survey attention check would be helpful to other researchers involved in similar studies. Perhaps you could include this explanation as a footnote.

To me the term 'wave' in a longitudinal study implies repeated measures on the same sample. Perhaps you could call your surveys 'Time1' and 'Time 2' to avoid any possible confusion.

	Abstract, line 6: introduced in 2018/Introduced in Colombia in 2018
REVIEWER	Ravara, Sofia University of Beira Interior (UBI), Health Sciences Research Centre (CICS), Faculty of Health Sciences (FCS)
REVIEW RETURNED	28-Oct-2021
GENERAL COMMENTS	This is a relevant and well-performed observational study exploring the impact of a new set of pictorial warnings in Colombia smokers. It is important to assess the impact of this tobacco control policy in different countries, since design features and its implementation vary widely. There are a few but important major issues to discuss.  1. On the methods section, authors classify the study as a longitudinal online survey with 2 waves of data collection; actually, the study is not a longitudinal study but a cross-sectional study performed in two waves (pre-post implementation of the pictorials) and using two different samples of smokers. While we understand that it is more difficult to perform cohort studies than cross-sectional studies, the classification of the study should be reviewed. 2. In the results section, authors should clarify what was the % of participants of the first wave that have collaborated in the second wave. This is not explicit in the manuscript. 3. In the results section, table 1, age should be depicted by confidence intervals. This would give a better picture of the participants' age distribution. 4. Since authors have gathered information on sociodemographic variables such as age and sex, it would be relevant to explore any study outcomes differences by sex or age. The abstract should be modified according to the reviewer comments.

VERSION 1 – AUTHOR RESPONSE

Reviewer: 1

Your surveys were conducted online. I believe the level of internet penetration in Colombia is only 65%, so there is potential coverage error in your samples. Also, your convenience sampling methods are likely to have produced non-representative samples of those with access to the internet. Consequently, it is not clear how representative your samples were of Colombian smokers in general (for example, I suspect your respondents would be better educated than the general population). Ideally, you would include in Table 1 the characteristics of Colombian smokers in general and comment on any significant differences between your samples and the smoker population.

We agree with the reviewer that our sample is unlikely to be fully representative of Colombian smokers and non-smokers. We have included this as a limitation in the discussion section on pages 17-18. There is a lack of published data on the demographics of Colombian smokers to enable the direct comparison with the data in Table 1 of our manuscript. However, we have included an indirect comparison of our data with data from one other study (Storr et al, 2008) in our discussion.

This comparison suggests that in our sample, smokers were on average younger, more likely to be male, and more educated than in previous reports.

I understand the logic of your pre-post design and appreciate that you have separate analyses for all of your wave 2 respondents and only for those who were not 'exposed' in wave 1. However, I wonder how much value there actually is in comparing your respondents' evaluations a year apart. The significant differences in mean scores in Tables 2 and 4 are almost all less than five percentage points and they are highly significant simply because of the very large sample sizes concerned. Arguably, many of these differences are not practically important, and your between-subjects design means it is difficult to know if these differences are 'real' or simply a reflection of differences in your samples.

We appreciate that the differences reported between data collection waves whilst statistically significant may not reflect meaningful differences in outcomes such as negative affect and cognitive elaboration. We have acknowledged this in our discussion on page 15.

These between-sample comparisons are necessary to test your 'wear-out' hypothesis, but as I explain below, I believe wear-out is only one plausible explanation for your findings and not one that can be proven by these findings. The deficiencies in the Colombian warnings are well illustrated by the results of your second survey and support your conclusions that something needs to be done to improve the notability, exposure and effectiveness of these warnings. However, if you want to continue to argue that your study is a test of 'wear out', I think you need to clearly explain why 'wear-out' is the inevitable cause of the reduction in negative effect, etc. in wave 2 and acknowledge that statistically significant differences may not be practically important. In other words, I think you should be a little more circumspect in the interpretation of your results.

Regardless of this, I am not sure your results support your conclusion that wear-out is a major cause of the reduced effectiveness of the new warnings you observed, or that larger warnings would reduce this wear-out effect. Colombian smokers may well have become desensitised to warnings about miscarriage, second hand smoke and heart attack. However, as I have already mentioned, your between-subjects design makes it difficult to be certain about the apparent decline in the effectiveness of these warnings or its causes and you do not explain how larger warnings would reduce wear-out (if this is what is occurring). Furthermore, I believe there are other equally plausible conclusions that could be drawn from your results.

As you point out, the Colombian warnings cover only 30% of the pack, not the 50% recommended by the WHO, so the warnings are relatively small and arguably easy to ignore or overlook. The images themselves seem rather benign to me, compared to some of the gruesome health images used in other countries. Also, four of the six images are unlikely to be very convincing in my opinion: the anxiety and pancreatic cancer images are not obviously associated with the consequences of smoking, and the death and second hand smoke images are very similar and not particularly compelling. I appreciate that the design and choice of warning images is not an easy task, but I think emphasising wear-out ignores the real problem with the Colombian warnings; namely, that they are too small and not well executed.

We have toned down "wear-out" as the only explanation of the differences observed between study waves. We have now included an alternative explanation, suggesting that the studied warnings may simply not be effective at influencing responses such as perceived effectiveness and negative affect, namely due to their size.

A few minor points:

A brief explanation of your survey attention check would be helpful to other researchers involved in similar studies. Perhaps you could include this explanation as a footnote.

This has now been added on page 9.

To me the term 'wave' in a longitudinal study implies repeated measures on the same sample. Perhaps you could call your surveys 'Time1' and 'Time 2' to avoid any possible confusion.

We have changed this throughout the manuscript.

Abstract, line 6: introduced in 2018/Introduced in Colombia in 2018

This has now been changed on page 2.

Reviewer: 2

On the methods section, authors classify the study as a longitudinal online survey with 2 waves of data collection; actually, the study is not a longitudinal study but a cross-sectional study performed in two waves (pre-post implementation of the pictorials) and using two different samples of smokers. While we understand that it is more difficult to perform cohort studies than cross-sectional studies, the classification of the study should be reviewed.

This has been changed throughout the manuscript.

In the results section, authors should clarify what was the % of participants of the first wave that have collaborated in the second wave. This is not explicit in the manuscript.

Unfortunately, it is not possible for us to determine whether any participants from the first wave also participated in the second wave due to the anonymity of survey.

In the results section, table 1, age should be depicted by confidence intervals. This would give a better picture of the participants' age distribution.

These have been added to Table 1.

Since authors have gathered information on sociodemographic variables such as age and sex, it would be relevant to explore any study outcomes differences by sex or age.

Unfortunately, we did not plan for these comparisons in our pre-registered protocol and therefore would not be sufficiently powered to conduct these analyses. These data were collected simply to provide demographic characteristics of our sample.

VERSION 2 – REVIEW

REVIEWER	Gendall, Philip University of Otago, Public Health
REVIEW RETURNED	20-Feb-2022
GENERAL COMMENTS	Your paper presents an interesting analysis of the impact of the new tobacco health warnings in Colombia. However, despite what you say in your response to reviewers, you still conclude that 'wear

	out' is the main problem. "Our data indicate a pattern of "wear-out" in warning effectiveness. Tobacco control policy should seek to improve exposure to tobacco health warnings in Colombia and to reduce wear-out effects." Your study does seem to suggest that smokers have become inured to some common health warning themes, but this is not proof that the new on-pack warnings have 'worn out' (in less than 12 months). You have not distinguished between the warning themes and the on-pack warnings themselves. I am not suggesting that 'wear out' of some warning themes is not an issue, but in my view you give too much emphasis to this explanation for the poor performance of the new warnings, with little evidence that this is the main problem. For example, you argue that larger warnings would reduce wear out, when it seems more likely they would increase noticeability. In your abstract you say "Between waves, we observed a reduction in negative affect towards current warnings..". Since none of the respondents in the first wave could have seen the current warnings, I'm not sure how you can conclude this. I think your paper has something relevant and useful to say about on-pack tobacco health warnings, but in my view the value of this understanding this is obscured by an over-emphasis on 'wear out' and a disconnect between the discussion of your results and your conclusions.
--	--

REVIEWER	Ravara, Sofia University of Beira Interior (UBI), Health Sciences Research Centre (CICS), Faculty of Health Sciences (FCS)
REVIEW RETURNED	20-Feb-2022
GENERAL COMMENTS	Authors have fully adressed reviewers' concerns and comments. I have no further comments.

VERSION 2 – AUTHOR RESPONSE

Reviewer 2.

1. Your paper presents an interesting analysis of the impact of the new tobacco health warnings in Colombia. However, despite what you say in your response to reviewers, you still conclude that 'wear out' is the main problem. "Our data indicate a pattern of "wear-out" in warning effectiveness. Tobacco control policy should seek to improve exposure to tobacco health warnings in Colombia and to reduce wear-out effects."

I think your paper has something relevant and useful to say about on-pack tobacco health warnings, but in my view the value of this understanding this is obscured by an over-emphasis on 'wear out' and a disconnect between the discussion of your results and your conclusions.

We have amended the conclusion section in the abstract (page 2) and main text (page 18) to reflect the findings of our study, with a focus on improving exposure and noticeability to warnings rather than reducing wear-out.

Your study does seem to suggest that smokers have become inured to some common health warning themes, but this is not proof that the new on-pack warnings have 'worn out' (in less than 12 months). You have not distinguished between the warning themes and the on-pack warnings themselves. I am not suggesting that 'wear out' of some warning themes is not an issue, but in my view you give too much emphasis to this explanation for the poor performance of the new warnings, with little evidence that this is the main problem. For example, you argue that larger warnings would reduce wear out, when it seems more likely they would increase noticeability.

We have amended our discussion of wear-out as the main explanation for poor performance of the warnings (Page 15). For warnings depicting familiar themes we suggest that the diseases depicted may themselves be the cause of the reduced effectiveness we observe at both waves.

We have also amended our argument that larger warnings would reduce wear-out, rather focusing on the potential for larger warnings to improve noticeability and exposure (Page 17).

2. In your abstract you say "Between waves, we observed a reduction in negative affect towards current warnings..". Since none of the respondents in the first wave could have seen the current warnings, I'm not sure how you can conclude this.

We apologise if this was not clear, but participants viewed the six new warnings in the online survey at both wave 1 and 2. Participants were asked complete measures of negative affect, believability, thinking about harms, reactance and perceived message effectiveness in response to the new warnings at each wave. In the manuscript we have highlighted the fact that participants completed the same measures at each wave, which includes exposure to and rating of the six new warnings (Page 9, procedures section).